# DISCREPANCY-AWARE KNOWLEDGE DISTILLATION FOR LARGE LANGUAGE MODELS

## ABSTRACT

Knowledge Distillation (KD) is a key technique for enhancing the capabilities of student models by transferring knowledge from powerful teachers. In Large Language Models (LLMs), however, the effectiveness of this transfer is fundamentally limited by distributional mismatch. The generic data used for distillation often fails to reflect the specialized distribution underpinning core expertise of the teacher. This gap hinders the acquisition of the teacher's most valuable capabilities. The challenge is fundamental because the ideal corrective method, importance weighting, is intractable without access to the unknown target density. We propose **D**iscrepancy **A**ware **K**nowledge **D**istillation (DAKD), a framework that re-frames this problem. Instead of estimating the unknown distribution, DAKD approximates the ideal importance weights by measuring the predictive discrepancy between the full teacher and a pre-trained-only base teacher, which serves as a distributional probe. The DAKD framework is "discrepancy aware" in a dual sense. It leverages the teacher-base divergence for distributional correction while using the teacher-student divergence for adaptive learning focus. This re-weighting is applied across multiple granularities, from the sequence and position down to the vocabulary level. Extensive experiments show that DAKD substantially outperforms state-of-the-art methods, enabling student models to more effectively inherit the nuanced capabilities of more powerful teachers.

## 1 INTRODUCTION

Large language models (LLMs) have greatly advanced the field of natural language processing (NLP), achieving state-of-the-art performance across a wide range of tasks Floridi & Chiriatti (2020); Touvron et al. (2023). This success is largely due to their ability to scale both parameter size and pre-training data. However, these advancements come with substantial computational and memory costs Minaee et al. (2024), making it challenging to deploy LLMs in resource-constrained environments, such as edge devices or real-time applications. Knowledge distillation (KD), an effective model compression approach, has emerged as a practical solution, compressing large teacher models into smaller student models that maintain high performance while being more efficient and deployable Park et al. (2019).

However, the efficacy of KD hinges on a critical, often violated, assumption: that the distillation data distribution ($\mu_0$) mirrors the quality of the data that forged the teacher's expertise ($\pi$). In practice, a state-of-the-art teacher model $\mathcal{M}_s$ achieves its advanced capabilities through a qualitative leap from its base model using a premium and often proprietary data distribution ($\pi$). This alignment data, refined through resource-intensive processes like supervised fine-tuning (SFT) and reinforcement learning with human feedback (RLHF) Ouyang et al. (2022), embeds exceptionally valuable and inaccessible human knowledge. Distillation, conversely, typically relies on generic data ($\mu_0$) that lacks the same signal density. This creates a crucial distributional mismatch, biasing the student to merely replicate the teacher's surface-level predictions while failing to generalize the core, alignment-derived principles required for high performance.

To address this challenge, we propose a new paradigm that reframes the problem. Instead of pursuing the intractable task of modeling the absolute target distribution $\pi$, we focus on the practical goal of approximating the ideal importance weights that distinguish $\pi$ from $\mu_0$. Our core insight is to introduce a pre-trained-only **base teacher** ($\mathcal{M}_b$) to serve as a distributional probe. Crucially, the accessibility of this base model is not a theoretical convenience but a practical reality within the

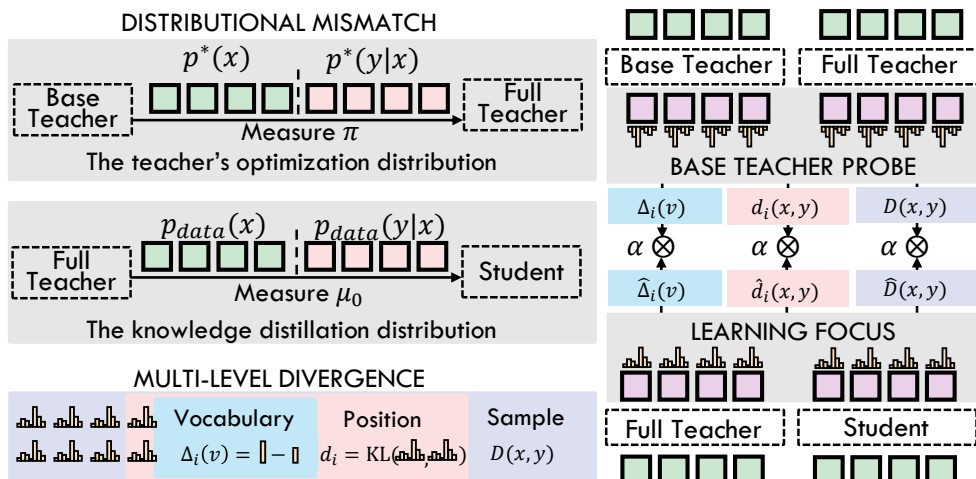

Figure 1: The Discrepancy Aware Knowledge Distillation (DAKD) Framework. The DAKD methodology is motivated by two key challenges (Top-Left): the Distributional Gap between empirical and ideal data, and the Knowledge Gap between the teacher and student. The core DAKD Engine (Right) addresses these via two parallel probes: a Distributional Probe, which computes the $\mathcal{M}_s - \mathcal{M}_b$ divergence to find relevant data, and a Learning Probe, which computes the $\mathcal{M}_s - \mathcal{M}_{stu}$ discrepancy to find challenging patterns. These signals are fused ($\alpha$) to produce the final output (Bottom-Left): multi-level Adaptive Weights, which are then used to modulate the standard KD loss.

modern LLM ecosystem. A prevailing trend in the release of state-of-the-art open-source models, as exemplified by prominent families such as Llama Touvron et al. (2023), Gemma Team et al. (2024b), and Mistral Jiang et al. (2023), is the concurrent provision of both the pre-trained base model and the final, finetuned version. This practice renders the base model a readily available yet largely underexploited resource for probing the effects of alignment, forming the cornerstone of our proposed methodology. Formally, $\mathcal{M}_b$ represents the teacher's state *prior* to alignment, from which the full teacher ($\mathcal{M}_s$) is fine-tuned. The predictive discrepancy between $\mathcal{M}_s$ and its predecessor therefore offers a powerful, observable proxy for the "alignment effect." A large divergence on a given sample suggests it is highly characteristic of the target distribution $\pi$, providing a principled mechanism to re-weight the available data and concentrate the distillation process on the most salient contexts.

Building on this principle, we introduce **D**iscrepancy **A**ware **K**nowledge **D**istillation (DAKD), a novel framework that operationalizes this insight. The DAKD framework is "discrepancy aware" in a dual sense. It first leverages the divergence between the full teacher and the base teacher for **distributional correction**. This is complemented by a second signal, the divergence between the teacher and the student, which provides an adaptive **learning focus**. This dual-signal re-weighting is applied comprehensively across multiple granularities, from the sequence and position down to the vocabulary level, to create a principled and dynamic learning curriculum.

Our primary contribution is the formalization of the distributional mismatch problem, stemming from a quality gap in data, as a key bottleneck in the distillation of modern language models. We then propose a novel and practical solution that introduces a base teacher as a distributional probe to approximate the ideal distillation distribution without requiring access to the teacher's proprietary alignment data. This principle is instantiated in our comprehensive DAKD framework, which employs the dual-signal, multi-level re-weighting scheme to create a highly adaptive distillation process. We validate our approach with extensive experiments, demonstrating that DAKD achieves state-of-the-art performance on a range of challenging tasks.

## 2 RELATED WORK

### 2.1 KNOWLEDGE DISTILLATION FOR LLMS

Knowledge Distillation (KD) Hinton (2015) is a traditional model compression technique that transfers knowledge and capabilities from a large model to a smaller student model Zhu et al. (2024).

In the field of large language model distillation, SeqKD Kim & Rush (2016) is considered a simple yet efficient distillation method for black-box LLMs. SeqKD finetunes the student model by maximizing the likelihood of sequences produced by the teacher model, On the other hand, for white-box teacher models Zhao & Zhu (2023), minimizing the divergence between the logits output by the teacher and those output by the student is a more intuitive distillation method Xu et al. (2024). Forward Kullback-Leibler Divergence (FKLD) is the primary method used in KD to measure the divergence between distributions Sanh (2019); Timiryasov & Tastet (2023). While MINILLM Gu et al. (2024) employs Reverse Kullback-Leibler Divergence (RKLD) to measure. DISTILLM Ko et al. (2024) adopts a mixed distribution and calculates skew KLD. Moreover, DAC-KL loss Liu et al. (2024b) is introduced to mitigate the interference of redundant information during distillation. On the other hand, Self-Evolution KD Song et al. (2024) addresses the issue of the student model's limited learning capacity by alleviating the difficulty of learning the distribution of hard-to-learn tokens. Nevertheless, the aforementionedmethods are built upon the assumption of distributional alignment, neglecting that discrepancies in data quality may hinder the effectiveness of distillation.

## 2.2 DISTRIBUTION LIMITATIONS IN LLM DISTILLATION

Typically, Knowledge Distillation requires training the student model on the same dataset used to train the teacher Ma et al. (2021); Liu et al. (2024a). For instance, DistilBERT Sanh (2019)performs distillation using BERT's original training corpus. However, in practical LLM distillation scenarios, privacy concerns and proprietary restrictions prevent access to the original training data Rashid et al. (2021). The training corpora of mainstream models (e.g., Gemma Team et al. (2024a), Mistral Jiang et al. (2023), and LLaMA Touvron et al. (2023)) are not publicly released, rendering the underlying distribution fundamentally inaccessible. Data-free knowledge distillation methods, such as PromptDFD Ma et al. (2022), similarly emphasize that, due to the unavailability of the original training distribution in LLMs, these methods reconstruct a synthetic dataset that approximates the target distribution to facilitate distillation.

## 2.3 DISCREPANCY-DRIVEN KNOWLEDGE DISTILLATION

The large discrepancy between the teacher and student models can degrade the effectiveness of knowledge distillation. Several discrepancy-driven distillation methods have been proposed to address this issue. TAKD Mirzadeh et al. (2020) mitigates the limited capacity of the student when the teacher is too complex by employing a two-stage distillation process. DIST Huang et al. (2022) tackles the problem of severe prediction discrepancies caused by an overly powerful teacher by using Pearson correlation instead of KL divergence. In the LLMs field, DenKD Ge et al. (2024) introduces a discrepancy-aware denoising representation learning approach to alleviate pseudo-label noise caused by the source-target language gap.

# 3 PRELIMINARIES

## 3.1 KNOWLEDGE DISTILLATION FOR LANGUAGE MODELS

Knowledge Distillation (KD) transfers knowledge from a large teacher model $\mathcal{M}_s$ to a smaller student model $\mathcal{M}_{stu}$ Hinton (2015). KD trains the student to match the teacher's soft prediction distributions over the vocabulary, providing a richer supervisory signal than hard labels.

In autoregressive language modeling, the models generate token sequences $\boldsymbol{y} = (y_1, \ldots, y_L)$ from a vocabulary $\mathcal{V}$, conditioned on context $\boldsymbol{x}$. At each step $i$, the model generates a distribution over the next token given the prefix $\boldsymbol{y}_{<i}$. The logits produced by the teacher and student are denoted $\mathbf{z}_i^s = \mathcal{M}_s(\boldsymbol{x}, \boldsymbol{y}_{<i})$ and $\mathbf{z}_i^{stu} = \mathcal{M}_{stu}(\boldsymbol{x}, \boldsymbol{y}_{<i})$, respectively, which are converted to probabilities using temperature-scaled softmax:

$$\mathbf{p}_i^s(\cdot; \tau) = \text{softmax}(\mathbf{z}_i^s/\tau), \quad \mathbf{p}_i^{stu}(\cdot; \tau) = \text{softmax}(\mathbf{z}_i^{stu}/\tau), \tag{1}$$

The standard KD objective minimizes the KL divergence between the teacher and student prediction distributions. The total loss is aggregated over a set of token positions $\mathcal{I}(\boldsymbol{x}, \boldsymbol{y})$ for each example in

the dataset $\mathcal{D}$:

$$\mathcal{L}_{\text{KD}}(\mathcal{M}_{stu}) = \mathbb{E}_{(\boldsymbol{x},\boldsymbol{y})\sim\hat{p}_{\text{data}}} \left[ \frac{1}{|\mathcal{I}(\boldsymbol{x},\boldsymbol{y})|} \sum_{i\in\mathcal{I}(\boldsymbol{x},\boldsymbol{y})} \text{KL}\big(\mathbf{p}_i^s \,\|\, \mathbf{p}_i^{stu}\big) \right], \tag{2}$$

## 3.2 THE DISTRIBUTIONAL MISMATCH PROBLEM

KD assumes that the distillation dataset $\mathcal{D}$ is representative of the data distribution under which the teacher was trained. Let this true training distribution be $p^\star$. Ideally, the student should be optimized under $p^\star$:

$$\mathcal{L}^\star(\mathcal{M}_{stu}) = \mathbb{E}_{(\boldsymbol{x},\boldsymbol{y})\sim p^\star} \left[ \frac{1}{|\mathcal{I}(\boldsymbol{x},\boldsymbol{y})|} \sum_{i\in\mathcal{I}(\boldsymbol{x},\boldsymbol{y})} \text{KL}\big(\mathbf{p}_i^s \,\|\, \mathbf{p}_i^{stu}\big) \right]. \tag{3}$$

However, the true data distribution $p^\star$ is often inaccessible, and we rely on a generic dataset $\mathcal{D}$, whose empirical distribution $\hat{p}_{\text{data}}$ may differ from $p^\star$. This discrepancy leads to two main challenges: **Query Shift** ($\hat{p}_{\text{data}}(\boldsymbol{x}) \neq p^\star(\boldsymbol{x})$): The input distribution in the distillation data differs from the teacher's domain. For example, a teacher fine-tuned on high-quality data ($p^\star$) might be distilled using generic web text ($\hat{p}_{\text{data}}$). **Trajectory Shift** ($\hat{p}_{\text{data}}(\boldsymbol{y}\mid\boldsymbol{x}) \neq p^\star(\boldsymbol{y}\mid\boldsymbol{x})$): Even for the same query $\boldsymbol{x}$, the reference sequences $\boldsymbol{y}$ in $\hat{p}_{\text{data}}$ may not reflect the teacher's style or reasoning quality.

This mismatch forces the student to imitate the teacher on out-of-distribution contexts, while failing to learn the teacher's behaviors in the target domain $p^\star$. A typical solution would be importance sampling, but estimating the density ratio is intractable in high-dimensional text sequences.

Our key insight is that while the ideal weight is latent, we can approximate it by observing predictive divergences between models. By incorporating these divergences into the distillation objective, we can effectively re-weight the training measure without needing to estimate the density ratio, mitigating the distributional mismatch and improving the distillation process.

## 4 METHOD

### 4.1 CORRECTING MISMATCH WITH A BASE TEACHER PROBE

Our methodology addresses the distributional mismatch between the empirical distillation measure $\mu_0$, and the teacher's ideal target measure $\pi$. This approach is grounded in the principle of *measure tilting*, formalized in Proposition 1. The key idea behind this principle is to construct a score function $s(u)$ that, when used to tilt the initial measure $\mu_0$, yields a distribution $\mu_\lambda$ that increasingly approximates the target distribution $\pi$ as $\lambda$ increases.

**Proposition 1** *Let $\mu_0$ be an initial probability measure and $\pi$ a target measure on a space $\mathcal{U}$. For a measurable score function $s : \mathcal{U} \to \mathbb{R}$, consider the exponentially tilted measure $\mu_\lambda$ for $\lambda \geq 0$, defined by*

$$d\mu_\lambda(u) \propto e^{\lambda s(u)} d\mu_0(u).$$

*If the score function satisfies*

$$\mathbb{E}_\pi[s] > \mathbb{E}_{\mu_0}[s],$$

*then the Kullback-Leibler divergence from $\pi$ to $\mu_\lambda$ is strictly decreasing at $\lambda = 0$:*

$$\frac{d}{d\lambda} KL(\pi\|\mu_\lambda)\bigg|_{\lambda=0} < 0.$$

Proposition 1 provides a clear directive: the task of correcting the distributional mismatch between $\mu_0$ and $\pi$ reduces to constructing an observable score function $s(u)$ that has a higher expectation under the target measure $\pi$ than under the initial measure $\mu_0$. While the optimal score is the intractable log-density ratio $s^\star(u) = \log\left(\frac{d\pi}{d\mu_0}(u)\right)$. To achieve this, we introduce a **base teacher** $\mathcal{M}_b$, with parameters $\theta_{\text{B}}$ obtained from pre-training, and a **full teacher** $\mathcal{M}_s$, with parameters $\theta_{\text{S}}$ obtained by fine-tuning $\mathcal{M}_b$ on the target distribution $\pi$, formally defined as:

$$\mathcal{M}_b = \arg\min_\theta \mathbb{E}_{u\sim\mu_{\text{pre}}}[\mathcal{L}_{\text{pretrain}}(\theta; u)], \quad \mathcal{M}_s \approx \arg\min_\theta \mathbb{E}_{u\sim\pi}[\mathcal{L}_{\text{align}}(\theta; u)] \text{ starting from } \theta_{\text{B}}. \tag{4}$$

We then define our score function as the KL divergence between their predictions as:

$$s(u) := \mathrm{KL}(p_{\mathcal{M}_s}(\cdot \mid u; \theta_{\mathrm{S}}) \,\|\, p_{\mathcal{M}_b}(\cdot \mid u; \theta_{\mathrm{B}})) . \tag{5}$$

The transition from $\theta_{\mathrm{B}}$ to $\theta_{\mathrm{S}}$ is driven by gradients of an alignment loss $\mathcal{L}_{\mathrm{align}}$ under the target distribution $\pi$. These gradients induce the largest parameter updates for instances $u$ that are most characteristic of the alignment task. Thus, the score $s(u)$ serves as a macroscopic measure of this predictive change. Since the instances that induce the largest predictive changes are more prevalent under $\pi$ than $\mu_0$, the score function $s(u)$ will, on average, yield higher values for samples from $\pi$, as these samples lead to larger updates during fine-tuning. Therefore, the score satisfies the condition in Proposition 1, ensuring that the distribution of $\mu_\lambda$ tilts towards $\pi$ as $\lambda$ increases.

## 4.2 MULTI-LEVEL DIVERGENCE AWARE KD

The challenge identified in Section 3.2 is that the distributional mismatch between $\hat{p}_{\mathrm{data}}$ and $p^\star$ is multi-faceted. This discrepancy arises not only from shifts in the marginal distribution over inputs $\boldsymbol{x}$ but also from shifts in the conditional distribution of sequences $\boldsymbol{y}$ given $\boldsymbol{x}$. A single correction at the sequence level is insufficient, as residual biases from uncorrected levels can dilute the training signal.

To address this, we propose a multi-level approach with scores derived from the divergence between the full teacher $\mathcal{M}_s$ and the base teacher $\mathcal{M}_b$. These scores are designed to correct the distributional mismatch at three granularities—sequence, position, and vocabulary—ensuring comprehensive alignment of the student model with the target distribution.

**Sequence-Level.** The sequence-level adjustment corrects the shift in the marginal input distribution, from $\hat{p}_{\mathrm{data}}$ to $p^\star$. This is accomplished by evaluating the alignment-relevance of an entire sequence, using the KL divergence between the full and base teacher distributions:

$$D(\boldsymbol{x}, \boldsymbol{y}) = \sum_i \mathrm{KL}\left(\mathbf{p}_i^s \,\|\, \mathbf{p}_i^b\right) , \tag{6}$$

where $\mathbf{p}_i^s$ and $\mathbf{p}_i^b$ represent the predictive distributions of the full teacher and the base teacher at position $i$, respectively. A high value of $D(\boldsymbol{x}, \boldsymbol{y})$ indicates that the context $(\boldsymbol{x}, \boldsymbol{y})$ elicits behaviors that strongly distinguish the aligned teacher from its pre-trained base model, signifying that the sequence is more characteristic of the target distribution $p^\star$.

The sequence-level score addresses **Query Shift**, where the distribution of input queries has changed. By correcting the overall input distribution mismatch, this score ensures the student model aligns with the global structure of the target task.

**Position-Level.** Within a sequence, not all positions carry equal importance. Key reasoning steps, factual answers, or stylistic markers are more indicative of the target conditional distribution inherent in $p^\star$ than filler language. To emphasize these critical regions, we introduce a position-specific score $d_i(\boldsymbol{x}, \boldsymbol{y})$ defined as the local KL divergence:

$$d_i(\boldsymbol{x}, \boldsymbol{y}) = \mathrm{KL}\left(\mathbf{p}_i^s \,\|\, \mathbf{p}_i^b\right) . \tag{7}$$

The position-level score addresses **Trajectory Shift**, focusing on specific positions where the model's behavior diverges from the base teacher. This allows the distillation process to focus on the most informative tokens, ensuring the gradients from key positions contribute more significantly to the student's learning.

**Vocabulary-Level.** Even within important positions, the supervisory signal can be spread across multiple tokens. The alignment effect often concentrates information on a few key vocabulary terms, shifting probability mass from a general synonym to a more precise technical term. To capture this, we define a vocabulary-level score $\Delta_i(v)$, based on the coordinate-wise probability discrepancy:

$$\Delta_i(v) = \log \mathbf{p}_i^s[v] - \log \mathbf{p}_i^b[v]. \tag{8}$$

This score quantifies the local information gain, identifying tokens where the teacher model's alignment most significantly alters the probability distribution. By focusing on these key tokens, the

score sharpens the learning signal and directs the student model's attention to the terms the teacher has learned to favor or disfavor during alignment.

The vocabulary-level score enhances the distillation process by emphasizing fine-grained changes in token probabilities. It prioritizes tokens that exhibit the most significant shifts in alignment knowledge, ensuring that the student model focuses on the most critical terms, leading to more accurate knowledge transfer.

## 4.3 UNIFIED SCORING AND FINAL OBJECTIVE

The teacher–base ($\mathcal{M}_s$–$\mathcal{M}_b$) divergence scores $D, d_i, \Delta_i$ serve the primary role of **distributional correction**. By reweighting training mass toward examples that most differentiate the aligned teacher from its base, they tilt the empirical measure $\mu_0$ toward the target $\pi$ (cf. Prop. 1).

A purely distributional tilt is static w.r.t. the student's progress. To introduce a **learning focus** that adapts to the student's state, we add student-aware counterparts ($\mathcal{M}_s$–$\mathcal{M}_{stu}$) using the same functional forms:

$$\hat{D}(\cdot) := \sum_i \mathrm{KL}\big(\mathbf{p}_i^s \,\|\, \mathbf{p}_i^{stu}\big), \quad \hat{d}_i(\cdot) := \mathrm{KL}\big(\mathbf{p}_i^s \,\|\, \mathbf{p}_i^{stu}\big), \quad \hat{\Delta}_i(v) := \log \mathbf{p}_i^s[v] - \log \mathbf{p}_i^b[v]. \tag{9}$$

**Combination.** To avoid ambiguity with the theoretical score $s(u)$ in Prop. 1, we denote the combined scores by $S$. At each level, we form a convex combination with $\alpha \in [0,1]$:

$$S_{\mathrm{pos}}(i) := (1-\alpha)\, d_i \,+\, \alpha\, \hat{d}_i, \quad S_{\mathrm{vocab}}(v \mid i) := (1-\alpha)\, \Delta_i(v) \,+\, \alpha\, \hat{\Delta}_i(v). \tag{10}$$

For the sequence-level score, we define the distributional term $D(x,y) = \sum_i \mathrm{KL}\big(\mathbf{p}_i^s \,\|\, \mathbf{p}_i^b\big)$ and the student-focus score $\hat{D}(x,y) = \sum_i \mathrm{KL}\big(\mathbf{p}_i^s \,\|\, \mathbf{p}_i^{stu}\big)$. To combine them, we introduce a de-centered version:

$$S_{\mathrm{seq}} = (1-\alpha)\, D(\boldsymbol{x},\boldsymbol{y}) + \alpha\, \widetilde{D}(\boldsymbol{x},\boldsymbol{y}), \tag{11}$$

where $\widetilde{D}(\boldsymbol{x},\boldsymbol{y})$ is the centered student score, defined as:

$$\widetilde{D}(\boldsymbol{x},\boldsymbol{y}) = \hat{D}(\boldsymbol{x},\boldsymbol{y}) - \frac{1}{|\mathcal{B}|} \sum_{(\boldsymbol{x}',\boldsymbol{y}') \in \mathcal{B}} \hat{D}(\boldsymbol{x}',\boldsymbol{y}'). \tag{12}$$

This centering ensures that the added term does not alter the first-order tilting direction of $D$, maintaining the measure tilt while introducing student adaptability. When $\lambda = 0$, the softmax normalization yields the standard exponential tilt based on $S_{\mathrm{seq}}$.

These raw scores are then converted into mean-1 normalized weights using a temperature $\lambda$, which controls the re-weighting strength:

$$w_{\mathrm{seq}} = B \cdot \mathrm{softmax}\big(\lambda S_{\mathrm{seq}}\big), \quad w_{\mathrm{pos}}(i) = m \cdot \mathrm{softmax}\big(\lambda S_{\mathrm{pos}}(i)\big),$$
$$w_{\mathrm{vocab}}(v \mid i) = |\mathcal{V}| \cdot \mathrm{softmax}\big(\lambda S_{\mathrm{vocab}}(v \mid i)\big), \tag{13}$$

where $B$ is the batch size and $m = |\mathcal{I}(\boldsymbol{x},\boldsymbol{y})|$. This softmax normalization is the finite-sample analog of exponential tilting on the corresponding product spaces (sequence / positions / coordinates), and yields unit-mean weights per level. This culminates in our final Discrepancy Aware Knowledge Distillation (DAKD) objective:

$$\mathcal{L}_{\mathrm{DAKD}} = \mathbb{E}_{(x,y) \sim \hat{p}_{\mathrm{data}}} \left[ w_{\mathrm{seq}} \sum_{i \in \mathcal{I}(x,y)} w_{\mathrm{pos}}(i) \sum_{v \in \mathcal{V}} w_{\mathrm{vocab}}(v \mid i)\, \mathbf{p}_i^s[v] \log \frac{\mathbf{p}_i^s[v]}{\mathbf{p}_i^{stu}[v]} \right]. \tag{14}$$

When $\lambda = 0$, all weights equal 1 and $\mathcal{L}_{\mathrm{DAKD}}$ reduces to standard KD. In implementation we treat the weights as constants w.r.t. student parameters (stop-gradient) to avoid biasing the KD gradient.

DAKD is *dual discrepancy aware*: the $\mathcal{M}_s$–$\mathcal{M}_b$ KL terms perform measure correction (guiding the student to the $\pi$-relevant regions), while the $\mathcal{M}_s$–$\mathcal{M}_{stu}$ terms concentrate learning on the student's current weaknesses without altering the first-order sequence-level tilt guaranteed by Prop. 1.

# 5 EXPERIMENTS

## 5.1 EXPERIMENTAL SETUP

We conducted a variety of experiments across two scenarios to validate the effectiveness of the proposed method. Following previous works Gu et al. (2024); Wu et al. (2024); Song et al. (2024), we firstly compared different distillation methods under the setting of same knowledge source, which means the teacher model and student models are pretrained with same corpora. Additionally, we conduct extra experiments on models with different knowledge source. In this scenario, although the models had the similar architecture, they were trained with different pretraining corpora.

**Base Models.** We utilized different sizes or generations of two series of models: for distillation experiments with the same knowledge source, we used GPT-2 Radford et al. (2019) 1.5B as the teacher model, and smaller variants (120M, 340M, 760M) as student models. For distillation experiments with different knowledge sources, Gemma2-9B-it and Gemma2-9B Team et al. (2024b) served as the full and base teachers, while Gemma-2B Team et al. (2024a) acted as the student. Additionally, Gemma2-2B-it and Gemma2-2B were employed as the full and base teachers for the Language Understanding and Reasoning tasks.

**Evaluations.** We follow the setting of Gu et al. (2024); Wu et al. (2024) and conducted experiments on five different instruction following datasets: Dolly Evaluation, Self-Instruct Wang et al. (2022a), Vicuna evaluation Chiang et al. (2023), Super-Natural Instructions Wang et al. (2022b), and Unnatural Instructions Honovich et al. (2022). Different from the setting in previous work, we adopt the greedy decoding strategy in the inference evaluation stage. This strategy can achieve more stable and even better results in terms of the ROUGE-L evaluation metric compared to the sampling decoding strategy. Additionally, we conduct experiments on three language understanding and reasioning tasks, which are Drop Dua et al. (2019), Hellaswag Zellers et al. (2019), and GSM8K Cobbe et al. (2021) to compare the effects of different knowledge distillation methods on the models' language understanding and reasoning abilities. In this part, we adopt Accuracy as the evaluation metric.

**Training.** For the scenario where Gemma family models, we do not train the teacher and directly let the teacher model generate responses for the instructions of different datasets to form the distillation training set. For the GPT-2 series, we follow the setting of Gu et al. (2024), splitting 14K samples from databricks-dolly-15k as the training set to first train the teacher model, and then use the fine-tuned model to generate answers to construct the distillation training set.

**Baselines.** We compare standard Cross-Entropy loss (SFT) and KL-divergence loss (SeqKD), along with several advanced distillation baselines, including AKL Wu et al. (2024) and Self-Evolution KD Song et al. (2024).

## 5.2 MAIN RESULTS

**Same Knowledge Source.** Table 1 shows results on different instruction-following tasks of student GPT2-120M learn from teacher GPT2-xl (1.5B) with different KD methods. The results show that DAKD has achieved a significant improvement (+2.86% average score) compared to the baseline SeqKD. When compared to other advanced baseline methods, some methods have achieved comparable results on certain tasks, but DAKD performs the best in most of tasks and in terms of average results.

**Different Knowledge Source.** To explore situations where there are greater discrepancies between the teacher and student, we use Gemma2-9B-it as the full teacher model, Gemma-2B as the student model and Gemma2-9B as the base teacher model. The results on the instruction-following evaluation datasets are presented in Table 2. We observe that the student model distilled through DAKD surpasses the baselines in all evaluation tasks and has the highest Average value. In particular, for S-NI, the result of DAKD outperform that of AKL, which achieved the second-best results, by 2.74%.

**Language Understanding and Reasoning Tasks.** In addition to testing on the tasks of instruction following, additional tasks targeting the language understanding and reasoning abilities of the models are introduced for comprehensive evaluation. Table 3 shows results on language understanding and reasoning tasks evaluated with OpenCompass. Given that the student model and the teacher model are pre-trained based on different corpora, the performance of our method outperforms that

Table 1: Main results of student GPT-2-120M on instruction-following tasks with different distilling methods. The best result is marked in **bold**, and the second best result is marked with an underline.

| Method | Dolly | SelfInst | Vicuna | S-NI | UnNI | Average |
|---|---|---|---|---|---|---|
| Teacher | 30.26 | 15.99 | 17.20 | 32.11 | 37.12 | 26.54 |
| SFT | 25.15±0.28 | 11.24±0.09 | 12.42±0.16 | 16.80±0.23 | 20.15±0.12 | 17.15 |
| SeqKD | 24.98±0.14 | 11.32±0.05 | 13.26±0.21 | 17.92±0.18 | 20.32±0.17 | 17.56 |
| SelfEvol | 24.32±0.19 | 10.83±0.11 | 12.85±0.24 | 20.36±0.11 | 25.15±0.26 | 18.70 |
| AKL | **26.12**±0.26 | 12.00±0.22 | 13.62±0.13 | 21.57±0.17 | 21.74±0.18 | 19.01 |
| DAKD | 26.03±0.12 | **12.36**±0.16 | **14.65**±0.09 | **22.69**±0.21 | **26.28**±0.24 | **20.40** |

Table 2: Main results of student Gemma-2b on instruction-following tasks with different distilling methods. The best result is marked in **bold**, and the second best result is marked with an underline.

| Method | Dolly | SelfInst | Vicuna | S-NI | UnNI | Average |
|---|---|---|---|---|---|---|
| Teacher | 26.05 | 23.48 | 24.9 | 44.48 | 36.65 | 31.11 |
| SFT | 23.27±0.19 | 17.75±0.08 | 22.04±0.24 | 35.86±0.17 | 33.57±0.11 | 26.50 |
| SeqKD | 23.94±0.05 | 19.35±0.17 | 22.15±0.29 | 38.45±0.08 | 34.63±0.23 | 27.70 |
| SelfEvol | 24.38±0.11 | 18.93±0.13 | 23.43±0.16 | 39.65±0.25 | 34.41±0.07 | 28.16 |
| AKL | 24.24±0.28 | 20.10±0.23 | 23.48±0.16 | 40.86±0.19 | 34.86±0.14 | 28.71 |
| DAKD | **25.84**±0.16 | **22.62**±0.16 | **24.37**±0.09 | **43.60**±0.06 | **36.23**±0.11 | **30.53** |

of other baseline methods across various evaluation datasets. We also observe that upon examining the HellaSwag dataset, the distilled student models generally outperform the teacher model, and we conduct a detailed analysis of the output differences between the teacher and student models. It is found that the teacher model's outputs are more diverse, which occasionally lead to the post-processing scripts being unable to accurately parse the correct options. In contrast, the student models' outputs, after undergoing knowledge distillation, became more uniform and could be correctly parsed in their entirety.

**Generalizability to Cross-Tokenizer Distillation.** To validate that DAKD can be applied in cross-tokenizer settings, we conduct a distillation experiment where the teacher model and the student model use different tokenizers. Specifically, the teacher model and the base model are Llama2-7b-chat and Llama2-7b, while the student model is Gemma-2b. We use ULD Boizard et al. (2024) to align the logits, which enables DAKD to operate in the cross-tokenizer scenario. The results presented in Table 4 show that DAKD consistently improves ULD, further supporting its generality and compatibility with existing cross-tokenizer distillation approaches.

## 5.3 ANALYSIS

**Comparison under Equal Training Cost.** We justify the additional inference cost by showing that DAKD delivers a higher return on compute compared to data expansion. We use Gemma2-2B-IT as the full teacher and Gemma2-2B as the base model, with Gemma-2B serving as the student. As a competitive strategy, we employ a data-expansion strategy that rephrases existing samples. Although the overhead introduced by DAKD is substantially smaller than the cost of generating 50% new data (around 20%), we conduct an even more stringent comparison. Table 5 reports results against baselines trained with an additional 50% generated data.

The empirical results indicate that DAKD consistently outperforms all computationally equivalent baselines. Even when the baselines are given a 50% larger data budget, they still fail to match DAKD's performance. This demonstrates that the extra inference pass in DAKD is not a wasteful cost but an efficient computational investment, fully justifying the incurred overhead.

**Data Efficiency in KD.** Knowledge distillation of LLMs often lacks sufficient high-quality training data, making data efficiency — achieving great performance with limited data — essential. To demonstrate the data efficiency of our method, we analyze the effectiveness of the proposed method and SeqKD on the GMS8K task, with the results presented in the left of Figure 2. The performance gap compared to the baseline gradually increases as the percentage of data used decreases. For

Table 3: Accuracy results of student Gemma-2b on language understanding and reasoning tasks.

| Method | Drop | GSM8K | Hellaswag | Average |
|--------|------|-------|-----------|---------|
| Full teacher | 39.96 | 52.84 | 56.20 | 49.67 |
| SFT | 39.52 | 40.07 | 51.93 | 43.84 |
| SeqKD | 39.53 | 43.82 | 53.24 | 45.53 |
| SelfEvol | 39.63 | 43.44 | 53.38 | 45.48 |
| AKL | 40.02 | 44.01 | 53.84 | 45.96 |
| DAKD | **40.83** | **45.39** | **55.95** | **47.39** |

example, when only 20% of the training data is used, our method achieves an accuracy of 39.03, while SeqKD achieves 34.37.

**Exploration of Student Model Size.** In general, as the size of the student model decreases, the distillation performance tends to worsen. This is because smaller student models have weaker capacity to match the teacher's distribution. To demonstrate that our method can mitigate the distillation difficulties caused by this, we conducted experiments with student models of different sizes. Specifically, GPT-2-1.5B is employed as the teacher model, and we selected three other sizes of GPT-2 (120M, 340M, 760M) as the student models. We present the average ROUGE-L in right of Figure 2. As student size decreases, both our method and SeqKD show performance drops, but ours declines more slowly, benefiting from alleviating the distributional mismatch problem during distillation process. When GPT-2-760M serves as the student model, our method achieves an improvement of 1.73% over SeqKD. With GPT-2-120M as the student model, the improvement increases to 2.94%.

Table 4: Accuracy results of student Gemma-2b on the Cross-tokenizer setting.

| Method | Dolly | SelfInst | Vicuna | S-NI | UnNI | Average |
|--------|-------|----------|--------|------|------|---------|
| ULD | 22.37 | 15.24 | 25.63 | 25.69 | 28.09 | 23.40 |
| ULD+DAKD | **22.89** | **17.82** | **26.81** | **27.10** | **31.76** | **25.28** |

Table 5: Comparison between DAKD and data-expansion baselines.

| Method | Data Volume | Dolly | SelfInst | Vicuna | S-NI | UnNI | Average |
|--------|-------------|-------|----------|--------|------|------|---------|
| SeqKD | 100% + 50%Gen. | 21.80 | 17.86 | 21.91 | 35.18 | 33.08 | 25.97 |
| SelfEvol | 100% + 50%Gen. | 22.34 | 17.56 | 21.73 | 36.65 | 32.29 | 26.11 |
| DAKD | 100% | **23.41** | **18.20** | **23.31** | **40.36** | **35.68** | **28.19** |

## 5.4 ABLATION STUDY

**Multi-Level Weighting.** To validate our hierarchical re-weighting design, we conduct an incremental ablation study, with results presented in Table 6. We begin with a standard KD baseline, which achieves a score of 43.82 on GSM8K. By introducing sequence-level weighting alone, which corrects for the overall distributional mismatch, performance immediately improves across all benchmarks, confirming the benefit of guiding the student to the teacher's target domain. Building upon this, the inclusion of position-level weights yields a further consistent performance boost, for instance, increasing the Drop score from 40.02 to 40.45. This demonstrates the value of concentrating the learning signal on the most salient tokens within a sequence. Finally, activating the vocabulary-level weights, which sharpens supervision on the most discriminative coordinates, achieves the best performance across all tasks, reaching 45.39 on GSM8K. These results strongly support our central design principle: a multi-level re-weighting strategy that corrects distributional shifts at progressively finer granularities is crucial for maximizing distillation effectiveness.

**Analysis of Components via Hyperparameter $\alpha$.** To rigorously evaluate the individual contributions of our two discrepancy signals and analyze their interplay, we present a comprehensive study by varying their balancing hyperparameter, $\alpha$. Figure 3 plots the performance on GSM8K against $\alpha$, serving as both a sensitivity study and a direct ablation study of our core components.

Table 6: Ablation analysis of the multi-level re-weighting components in DAKD.

| Method | Active Weighting Levels | | | Accuracy | | |
|---|---|---|---|---|---|---|
| | Sequence | Position | Vocabulary | GSM8K | Hellaswag | Drop |
| Standard KD (Baseline) | X | X | X | 43.82 | 54.52 | 39.53 |
| + Sequence-level | ✓ | X | X | 44.53 | 55.01 | 40.02 |
| + Position-level | ✓ | ✓ | X | 44.97 | 55.48 | 40.45 |
| + Vocabulary-level (Full DAKD) | ✓ | ✓ | ✓ | **45.39** | **55.95** | **40.83** |

First, we examine the ablation results at the extremes. At $\alpha = 0$, the model relies solely on the distributional correction signal ($\mathcal{M}_s - \mathcal{M}_b$). It achieves an accuracy of 45.02%, significantly outperforming the Standard KD baseline (43.82%). This result confirms the substantial, independent value of using the base teacher as a distributional probe. Conversely, at $\alpha = 1$, the model uses only the learning focus signal ($\mathcal{M}_s - \mathcal{M}_{stu}$) and achieves an accuracy of 44.16%, again clearly superior to the baseline. These endpoint results validate that both signal components are independently effective.

Crucially, the plot reveals a strong synergistic effect. The performance peaks at an optimal balance of $\alpha = 0.5$, where the full DAKD model achieves our best score of 45.39%. This result is substantially higher than that achieved by either individual component, demonstrating that the two signals are complementary and their balanced integration is key to maximizing performance. The overall unimodal trend validates our dual-signal design, confirming that both distributional correction and learning focus are vital, and their synergistic combination drives the success of the DAKD framework.

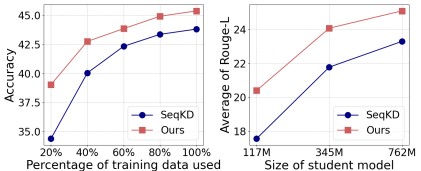

Figure 2: Left: GSM8K performance under different training sizes. Right: Avg. Rouge-L with GPT-2 students of different scales.

Figure 3: Results on GSM8K. Left: Accuracy with different $\alpha$ values. Right: Accuracy with different $\lambda$ values.

**Sensitivity to Re-weighting Strength $\lambda$.** We analyze the impact of the re-weighting strength $\lambda$, which controls the sharpness of the weights. With $\alpha$ fixed at its optimal value, we vary $\lambda$ and report the results on GSM8K in Figure 3. The plot shows a clear unimodal trend, confirming the importance of our re-weighting mechanism. All tested configurations of DAKD significantly outperform the SeqKD baseline. Performance rises as $\lambda$ increases from 1 to 5, peaking at $\lambda = 5$ with our best accuracy of 45.39%. This demonstrates that a sufficiently strong re-weighting is crucial for guiding the student model effectively. However, when $\lambda$ is increased further to 10, performance begins to degrade to 44.5%. This suggests that an overly aggressive re-weighting can be detrimental, likely by making the training unstable or by forcing the model to ignore valuable information in lower-scored instances. This analysis confirms that the strength of the DAKD re-weighting is a key, well-behaved hyperparameter that is optimal within a discernible range.

## 6 CONCLUSION

In this work, we introduced Discrepancy Aware Knowledge Distillation (DAKD), a novel framework that addresses the critical challenge of distributional mismatch in the distillation of Large Language Models. Our core mechanism is the introduction of a base teacher as a distributional probe. This enables a dual-signal, multi-level re-weighting scheme that corrects for the distributional gap via the teacher-base discrepancy while adaptively focusing on student weaknesses via the teacher-student discrepancy. By creating a more principled and adaptive training curriculum, DAKD enables student models to more effectively inherit the sophisticated, alignment-derived capabilities of their more powerful teachers, charting a new path toward data-efficient knowledge transfer.

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

## A  APPENDIX

This appendix contains

- Extended experimental results on instruction-following tasks;
- Additional experiment on the alignment task;
- Description of models employed as teachers and students;
- An empirical evaluation of training performance in terms of speed;
- Additional ablation results on weighting levels;
- Analysis of $\lambda$-scheduling: fixed optimal vs. annealed;
- Experiments on two alternative KL divergence formulations;
- Visualization of weighting mechanism;
- An overview of the baselines used in our experiments;
- The algorithm of the proposed Discrepancy-Aware Knowledge Distillation;
- Details regarding the application of large language models.

### A.1  EXTENDED EXPERIMENTAL RESULTS

To further demonstrate the general applicability of our method, we conduct additional experiments across different model settings beyond those in Section 5. Specifically, we retain the Gemma-2B model as the student and introduce a new configuration using Gemma2-2B-it as the full teacher model and Gemma2-2B as the base teacher model. These settings differ from our previous setup and serve to test the robustness. The results on instruction-following evaluation datasets are reported in Table A1. Across these settings, our proposed method DAKD continues to outperform the baseline methods, confirming its effectiveness in a broader range of scenarios.

Table A1: Main results of student Gemma-2b on instruction-following tasks with different distilling methods. The best result is marked in **bold**, and the second best result is marked with an underline.

| Method | Dolly | SelfInst | Vicuna | S-NI | UnNI | Average |
|---|---|---|---|---|---|---|
| Teacher | 24.37 | 22.53 | 24.45 | 43.26 | 33.86 | 29.69 |
| SFT | 21.37±0.14 | 16.70±0.08 | 17.31±0.23 | 31.06±0.19 | 32.48±0.05 | 23.78 |
| SeqKD | 22.40±0.06 | 17.27±0.18 | 22.32±0.27 | 35.65±0.09 | 34.16±0.22 | 26.36 |
| SelfEvol | 22.74±0.11 | 17.73±0.13 | 22.45±0.16 | 36.84±0.25 | 33.42±0.07 | 26.64 |
| AKL | 23.24±0.29 | 18.82±0.24 | 23.50±0.13 | 40.10±0.17 | 35.12±0.14 | 28.16 |
| DAKD | **24.34**±0.17 | **20.23**±0.18 | **23.91**±0.11 | **42.66**±0.08 | **35.65**±0.12 | **29.36** |

### A.2  ADDITIONAL EXPERIMENT ON THE ALIGNMENT BENCHMARK.

To demonstrate the method's breadth, we supplemented our evaluation with AlpacaEval[4], which is the standard benchmark for alignment. We conducted this experiment using Gemma2-2B-IT as the full teacher and Gemma2-2B as the base model, with Gemma-2B serving as the student. Specifically,

we employed GPT-4-1106-preview as the judge to perform pairwise comparisons between each method and the teacher model to calculate the win rate. The results shown in Table A2 demonstrate that DAKD outperforms the baseline and other comparative methods, confirming that it effectively captures the teacher's preference alignment.

Table A2: Results of Gemma-2B on AlpacaEval.

| Method | SFT | SeqKD | AKL | DAKD(Ours) |
|---|---|---|---|---|
| Win Rates ↑ | 11.64 | 11.90 | 11.27 | **14.57** |
| Std Error ↓ | 1.15 | 1.14 | **1.12** | 1.18 |

### A.3 DESCRIPTION OF MODELS

In Section 5 and Appendix A.1, Gemma-2-9B and Gemma-2-2B were used as teacher models, while Gemma-2B was used as the student model. In this experimental setup, the teacher model and the student model have different model architectures. We present the detailed information about the models in Table A3.

Table A3: Summary of the key model parameters.

| Model | # params | d_model | Layers | Num heads |
|---|---|---|---|---|
| gemma-2B | 2B | 2048 | 18 | 8 |
| gemma-2-2B | 2B | 2304 | 26 | 8 |
| gemma-2-9B | 9B | 3584 | 42 | 16 |

### A.4 TRAINING SPEED

To evaluate training speed, we follow the experimental setup of Different Knowledge Source in Section 5, using Gemma family models. All results are obtained using four A800 GPUs and presented in Figure A1. Although our method DAKD introduces additional computational cost, resulting in a slowdown of 1.9x compared to the vanilla knowledge distillation (SeqKD) method, it does not exhibit a significant disadvantage in training speed when compared to other distillation baseline methods, which experience slowdowns ranging from 1.5x to 1.7x. Considering the accuracy results from our experimental analysis, our method demonstrates superior model performance compared to other methods despite its slight speed reduction.

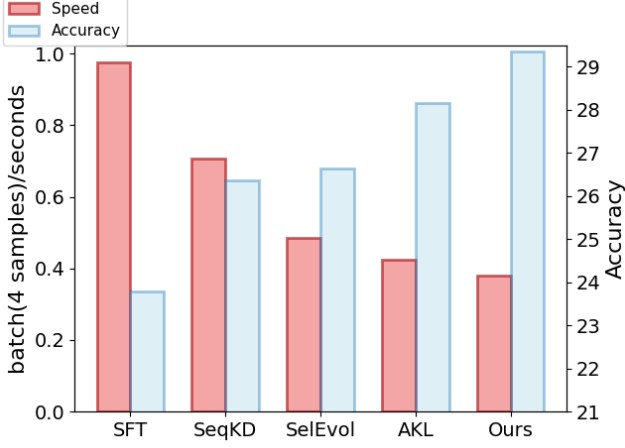

Figure A1: Training speed and experimental results of different methods

Table A4: Analysis of the independent contribution of each re-weighting level in DAKD. Each level is tested in isolation against the baseline, with the full model shown for comparison to highlight synergistic effects.

| Method | Active Weighting Levels | | | Accuracy | | |
|---|---|---|---|---|---|---|
| | Sequence | Position | Vocabulary | GSM8K | Hellaswag | Drop |
| Standard KD (Baseline) | X | X | X | 43.82 | 54.52 | 39.53 |
| DAKD (Sequence-level only) | ✓ | X | X | 44.53 | 55.01 | 40.02 |
| DAKD (Position-level only) | X | ✓ | X | 44.71 | 55.23 | 40.15 |
| DAKD (Vocabulary-level only) | X | X | ✓ | 44.05 | 54.80 | 39.81 |
| DAKD (Full, All levels) | ✓ | ✓ | ✓ | **45.39** | **55.95** | **40.83** |

## A.5 SUPPLEMENTARY ABLATION: INDEPENDENT CONTRIBUTION OF WEIGHTING LEVELS

To further dissect our multi-level weighting mechanism, we present a supplementary ablation study in Table A4 that evaluates the performance of each weighting level when applied in isolation. This analysis helps to understand the standalone contribution of each component.

The results indicate that both sequence-level and position-level weighting, when used individually, provide substantial improvements over the standard KD baseline. Position-level weighting appears to be the single most effective component, yielding a 0.89-point gain on GSM8K. This is intuitive, as it directly focuses the model on the most semantically critical parts of a sequence, such as the steps in a reasoning chain or the final answer. Sequence-level weighting also offers an independent gain, confirming the importance of up-weighting entire samples that are characteristic of the teacher's target distribution.

Interestingly, vocabulary-level weighting provides a more modest gain when used in isolation. This suggests its primary role is not to act as a standalone signal, but to refine and sharpen the supervisory signals provided by the coarser-grained levels. Crucially, the full DAKD model, which synergistically combines all three levels, markedly outperforms any single component. This demonstrates that while each level of weighting is independently beneficial, their true power is realized in their combination, where coarser levels identify the correct learning context and finer levels sharpen the focus within that context.

## A.6 COMPARISON BETWEEN FIXED OPTIMAL $\lambda$ AND THE ANNEALING STRATEGY.

We conducted a new experiment to evaluate the scheduling strategy for the hyperparameter Re-weighting Strength $\lambda$. We implemented a linear warm-up schedule for $\lambda$, increasing from 0 to 5 over the first 50% of training steps, and remaining fixed at 5 thereafter. We conducted this experiment using Gemma2-2B-IT as the full teacher and Gemma2-2B-Base as the base model, with Gemma-2B serving as the student. Table A5 shows that while the annealed $\lambda$ provides performance comparable to our optimal fixed $\lambda$, it does not offer additional gains. It indicates that while adaptive strategies are a valid design direction, DAKD is robust to the specific scheduling of $\lambda$ and performs effectively with a constant value.

Table A5: Comparison Between Fixed Optimal $\lambda$ and Annealing $\lambda$.

| Method | Drop | GSM8K | Hellaswag | Average |
|---|---|---|---|---|
| Fixed $\lambda = 5$ | **45.39** | **40.83** | **55.95** | **47.39** |
| Annealing $\lambda$ | 45.05 | 40.52 | 55.58 | 47.05 |

## A.7 ADDITIONAL EXPERIMENTS ON ALTERNATIVE KL DIVERGENCES.

We conduct additional experiments to verify the generality of DAKD. DAKD functions as a data importance re-weighting framework, which is orthogonal to the specific distance metric used for loss calculation. It can be seamlessly integrated with various divergences. We substitute the KL

divergence with both the $\alpha$-$\beta$-divergence and the $f$-divergence. We conducted this experiment using Gemma2-2B-IT as the full teacher and Gemma2-2B-Base as the base model, with Gemma-2B serving as the student. The corresponding experimental results are presented in Table A6. DAKD consistently improves performance regardless of the underlying divergence metric. This confirms that identifying and re-weighting alignment-relevant samples is a fundamental improvement that benefits various distillation objectives, verifying the method's robustness.

Table A6: Comparison between DAKD and data-expansion baselines.

| Divergence Metric | method | Dolly | SelfInst | Vicuna | S-NI | UnNI | Average |
|---|---|---|---|---|---|---|---|
| $f$-divergence(TVD) | w/o DAKD | 22.06 | 16.77 | 21.62 | 34.34 | 30.67 | 25.09 |
| $f$-divergence(TVD) | w DAKD (Ours) | **24.35** | **18.98** | **22.89** | **40.35** | **32.54** | **27.82** |
| $\alpha$-$\beta$-divergence | w/o DAKD | 22.25 | 17.55 | 22.67 | 36.01 | 30.92 | 25.88 |
| $\alpha$-$\beta$-divergence | w DAKD (Ours) | **23.50** | **19.37** | **23.48** | **41.94** | **33.18** | **28.30** |

### A.8 VISUALIZATION OF WEIGHTING MECHANISM

To demonstrate DAKD's dual-level weighting, we analyze the predictive probabilities and the resulting Importance Weights on two distinct samples: a Common Knowledge sample (Capital of France) and a Multi-step Reasoning sample (Arithmetic Calculation).

**Sample 1** (fact): What is the capital of France?

| Tokens | The | capital | of | France | is | Paris | Avg. |
|---|---|---|---|---|---|---|---|
| Base Model $p_{\texttt{base}}(y_i\|y_{<i})$ | 0.324 | 0.879 | 0.949 | 0.836 | 0.793 | 0.602 | **0.731** |
| SFT Model $p_{\texttt{sft}}(y_i\|y_{<i})$ | 0.424 | 0.945 | 0.988 | 0.984 | 0.156 | 0.953 | **0.742** |
| Weight | 0.81 | 0.68 | 0.60 | 1.03 | 0.02 | 2.86 | — |

**Sample 2** (reasoning): Calculate $8 + 4/2 =$?

| Tokens | Calculate | 4 | / | 2 | = | 2 |
|---|---|---|---|---|---|---|
| Base Model $p_{\texttt{base}}(y_i\|y_{<i})$ | 0.028 | 0.278 | 0.793 | 0.992 | 0.159 | 0.871 |
| SFT Model $p_{\texttt{sft}}(y_i\|y_{<i})$ | 0.34 | 0.984 | 0.992 | 1 | 0.238 | 0.879 |
| Weight | 0.55 | 3.95 | 0.31 | 0.12 | 0.17 | 0.12 |

| Tokens | 8 | + | 2 | = | 1 | 0 |
|---|---|---|---|---|---|---|
| Base Model $p_{\texttt{base}}(y_i\|y_{<i})$ | 0.605 | 0.941 | 0.98 | 0.578 | 0.087 | 0.996 |
| SFT Model $p_{\texttt{sft}}(y_i\|y_{<i})$ | 1 | 0.891 | 0.992 | 0.758 | 0.965 | 1 |
| Weight | 0.84 | 0.09 | 0.12 | 0.28 | 9.34 | 0.12 |

| Tokens | The | Answer | is | 1 | 0 | Avg. |
|---|---|---|---|---|---|---|
| Base Model $p_{\texttt{base}}(y_i\|y_{<i})$ | 0.048 | 0.017 | 0.816 | 0.957 | 0.988 | 0.596 |
| SFT Model $p_{\texttt{sft}}(y_i\|y_{<i})$ | 0.28 | 0.025 | 0.941 | 1 | 1 | 0.781 |
| Weight | 0.37 | 0.12 | 0.22 | 0.14 | 0.12 | — |

This table provides two clear, intuitive demonstrations:

**Sequence-Level Weighting (Filtering Data):** In Sample 1, the negligible probability gap ($\approx 0.01$) indicates the Base model already possesses this robust knowledge. Consequently, DAKD assigns a low global priority, preventing over-training on trivial facts. In addition, the significant gap signals shown in Sample 2 that the Base model struggles with the logic, while the teacher is confident. DAKD correctly identifies this as high-value alignment knowledge, triggering a high global weight.

**Position-Level:** In Sample 2, at token '4' (the operand), the weight jumps to 3.95 because the Base model (0.278) fails to identify the correct number to operate on, while the Teacher (0.984) is certain. Most notably, at the final answer token '1' (representing "10"), the weight peaks at 9.34. Here, the

Base model (0.087) completely fails the multi-step arithmetic, while the Teacher (0.965) derives the correct result.

This demonstration confirms that DAKD goes beyond simple text imitation. It automatically suppresses trivial tokens and heavily up-weights the specific reasoning steps (e.g., operands and results) where the teacher's alignment expertise provides the most critical correction to the base model.

### A.9   DISCTIPTION OF THE BASELINES

In the following, we provide a brief overview of all baseline methods used in Section 5.

- Cross-Entropy loss (SFT) performs supervised fine-tuning only on the student model, with cross-entropy loss as the loss function.
- SeqKD Kim & Rush (2016) fine-tunes the student model by aligning outputs with those of the teacher.
- AKL Wu et al. (2024) adaptively assigns weights to integrate Forward Kullback-Leibler Divergence and Reverse Kullback-Leibler Divergence.
- Self-Evolution KD Song et al. (2024) combines the teacher's distribution and the ground truth as prior knowledge in the KD process.

### A.10   ALGORITHM OF THE METHOD

To facilitate understanding of the training process of the proposed DA-KD, we present the complete training algorithm in Algorithm 1.

### A.11   THE USE OF LARGE LANGUAGE MODELS

This paper made use of large language models (LLMs) to polish the writing, including spelling corrections, minor sentence restructuring, and clarity enhancement. The authors confirm that all content modified by LLMs has been carefully reviewed to identify and eliminate potential errors, inaccuracies, or biases. The authors take full responsibility for the submitted paper and affirms that the work is original.

---

**Algorithm 1.** Discrepancy-Aware Knowledge Distillation (DAKD)

---

**Input:** $\mathcal{D}$: Distillation dataset, $\mathcal{M}_s$: Teacher model, $\mathcal{M}_b$: Base teacher model, $\mathcal{M}_{stu}$: Student model, $\alpha$: Hyperparameter for combining discrepancy signals, $\lambda$: Re-weighting strength

**Output:** $\mathcal{M}_{stu}$: Trained student model

**1 for** *each batch* $(x, y) \in \mathcal{D}$ **do**

    /* Compute teacher-base discrepancy for distributional
       correction                                              */
**2**   $D(x, y) \leftarrow \sum_i \text{KL}(\mathcal{M}_s(x, y < i) \| \mathcal{M}_b(x, y < i))$ ;
    /* Compute teacher-student discrepancy for learning focus  */
**3**   $\hat{D}(x, y) \leftarrow \sum_i \text{KL}(\mathcal{M}_s(x, y < i) \| \mathcal{M}_{stu}(x, y < i))$ ;
    /* Combine both signals to compute final weight at sequence
       level                                                    */
**4**   $w_{seq}(x, y) \leftarrow (1 - \alpha)D(x, y) + \alpha\hat{D}(x, y)$ ;
    /* Normalize sequence-level weights using softmax          */
**5**   $w_{seq}(x, y) \leftarrow \text{softmax}(\lambda \cdot w_{seq}(x, y))$ ;
**6**   **for** *each position* $i$ **do**
        /* Position-level re-weighting based on teacher-base and
           teacher-student discrepancy                          */
**7**       $w_{pos}(x, y, i) \leftarrow (1 - \alpha)\text{KL}(\mathcal{M}_s(x, y < i) \| \mathcal{M}_b(x, y < i)) + \alpha\text{KL}(\mathcal{M}_s(x, y < i) \| \mathcal{M}_{stu}(x, y < i))$;
        /* Normalize position-level weights using softmax       */
**8**       $w_{pos}(x, y, i) \leftarrow \text{softmax}(\lambda \cdot w_{pos}(x, y, i))$ ;
**9**   **end**
**10**  **for** *each vocabulary position* $v$ **do**
        /* Vocabulary-level re-weighting                        */
**11**      $w_{vocab}(x, y, i, v) \leftarrow |V| \cdot \text{softmax}(\lambda \cdot S_{vocab}(x, y, v))$ ;
**12**  **end**
    /* Final DAKD loss computation                              */
**13**  $\mathcal{L}_{DAKD} \leftarrow$
        $\mathbb{E}_{(x,y) \sim \mathcal{D}} \left[ w_{seq}(x, y) \sum_{i \in I(x,y)} w_{pos}(x, y, i) \sum_{v \in V} w_{vocab}(x, y, i, v) \log \frac{\mathcal{M}_s(x,y)[v]}{\mathcal{M}_{stu}(x,y)[v]} \right]$ ;
    /* Update student model with computed loss                  */
**14**  $\mathcal{M}_{stu} \leftarrow \mathcal{M}_{stu} - \text{grad}(\mathcal{L}_{DAKD})$
**15 end**

---

