# OpenReview forum: "Discrepancy-Aware Knowledge Distillation for Large Language Models"
_ICLR.cc/2026/Conference — Submitted to ICLR 2026_

### Official Review · Reviewer_DXgg · 2025-10-29

**Soundness:** 3
**Presentation:** 3
**Contribution:** 3
**Rating:** 4
**Confidence:** 3

**Summary:**

The authors propose DAKD, which essentially is a re-weighting of the standard KL divergence. Specifically, the author calculate the weights by considering at three levels: sequence, position, and vocabulary. The "importance" scores are calculated based on both the discrepancy between a finetuned teacher and a student, as well as between a finetuned teacher and pretrained teacher.

**Strengths:**

1. The proposed method show improvements over existing distillation methods.
2. The authors experimented with both instruction tuning and reasoning.

**Weaknesses:**

1. Higher performance could be due to hyper-parameter efforts. The authors combine the two discrepancies to produce these weights, which involves a alpha mixing hyper-parameter. As Table 3 shows, performance at 0 is pretty good, and as alpha goes to 1, half the values of alpha is better and the other half being worse.
2. Additional cost. The proposed method uses extra compute, because it requires an additional pass on the pretrained teacher model, which could be hard to justify if improvement is not significant.

**Questions:**

The paper states that " Ideally, the student would be optimized under the teacher’s true data distribution, p⋆... However, p⋆ is often inaccessible".

Why is this? We have the teacher's distribution and sampling from the teacher is a standard technique.

---

> ### Author Response · Authors · 2025-11-20
> **Response to Reviewer DXgg (1/3)**
>
> Dear Reviewer DXgg:
>
> We sincerely appreciate your constructive and thoughtful feedback.
>
> ## Q1. Analysis of the effect of the hyperparameter $\alpha$.
>
> We appreciate the careful reading of the $\alpha$ parameter (in Figure 3 Left). Here we clarify our method's robustness and synergistic design.
>
> **Every DAKD configuration ($\alpha \in [0, 1]$) outperforms the Standard KD baseline (43.82%)**. The fact that the combined result strictly exceeds either individual component ($\alpha=0$ or $1$) confirms the robust benefit of our dual-signal design.
>
> there may have been a slight misunderstanding regarding the baseline in Figure 3. The dashed line represents the Standard KD baseline (43.82%). When compared against this baseline, every single configuration of DAKD outperforms the standard method, regardless of $\alpha$.
>
> | **Configuration (α)** | **Meaning** | **GSM8K Accuracy** |
> | --- | --- | --- |
> | $\alpha = 0.0$ | Distributional Correction Only | 45.02% |
> | $\alpha = 1.0$ | Learning Focus Only | 44.16% |
> | $\alpha = 0.3$ | **Combined** | **45.39%** |
>
> This data highlights two crucial points that strongly support our design:
>
> 1. Method Robustness: As the table shows, all DAKD configurations, regardless of the $\alpha$ value (from 0.0 to 1.0), significantly outperform the standard baseline. This demonstrates that our method is robust and its gains are not due to fragile tuning.
> 2. Synergistic Design: As observation that the performance at $\alpha = 0.0$ is "pretty good" (45.02%) is insightful and correct. This result **independently validates** the core contribution of this work: using the $M_s−M_b$ discrepancy (Distributional Correction) as a distributional probe is effective on its own.
> Crucially, the performance *further improves* when combined with the student-aware signal (Learning Focus),which is the $M_s−M_{stu}$ term, peaking at 45.39% with $\alpha = 0.3$. This peak is higher than *either* individual component ($\alpha = 0.0$ or $\alpha = 1.0$), is not noise but rather a clear demonstration of synergy.
>
> In summary, the reviewer's analysis of $\alpha$ helps confirm that both components are independently valuable, and their combination provides an optimal and robust result.

---

> ### Author Response · Authors · 2025-11-20
> **Response to Reviewer DXgg (2/3)**
>
> ## Q2.  Additional cost: The proposed method uses extra compute, which could be hard to justify if improvement is not significant.
>
> We justify the additional inference cost by demonstrating that DAKD yields **a higher return on compute** than data expansion. Our analysis proves that DAKD (using 100% data) outperforms a baseline even when the baseline is allocated equivalent compute to generate and train on **50% more new data**.
>
> 1. **Simply increasing training epochs is ineffective.** First, allocating extra compute to train the student for more epochs on the *same* data is not a viable justification. Research [1] confirms that repeating tokens for LLMs yields diminishing returns and can lead to overfitting compared to learning from fresh data. Therefore, the only valid comparison is to convert the extra compute into generating and training on new data.
>
> 2. **Cost Equivalence Analysis**: To determine the fair amount of additional data $\Delta N$ that equates to the computational overhead of DAKD, we derived a cost equivalence based on the parameter ratio $R = \theta_{Tea} / \theta_{Stu}$. We assume a rigorous data expansion setting that includes both Data Generation ($1 \cdot \theta_{Tea}$) and Distillation ($1 \cdot \theta_{Tea}$ for teacher logits + $3 \cdot \theta_{Stu}$ for student training), resulting in a total per-sample cost of $2\theta_{Tea} + 3\theta_{Stu}$. Equating this to DAKD's overhead ($1 \cdot \theta_{Tea}$) yields the following exchange rate formula: $\Delta N = \frac{R}{2R + 3}$.
>
> We applied this formula to the model configurations used in our paper:
>
> - Scenario A (#$\theta_{Tea}=2B$ $\to $ #$\theta_{Stu}=2B$): With $R \approx 1$, the equivalent data increase is **20%.**
> - Scenario B (#$\theta_{Tea}=9B$ $\to$ #$\theta_{Stu}=2B$): With $R \approx 4.5$, the equivalent data increase is **37.5%**.
>
> Since our work involves multiple model scales, we adopted the upper bound (37.5%) as our reference. To conduct a stress test, we rounded this up to a **50% data increase** for the baseline. This means we granted the baseline significantly more compute resources than theoretically justified, yet DAKD still outperformed it, confirming the robustness of our method.
>
> 3. **Empirical Validation: DAKD (100%) > Baselines (150% Data).**
> To empirically verify this, we conducted a comprehensive comparison. We implemented two data expansion strategies for the baselines (SeqKD and SelfEvol) to reach **150% data volume.**
>
> - 150%: Additional 50% real Data.
> - 100% + 50% Gen: Expanding data via rephrasing existing samples by teacher model.
>
> We compared DAKD (using strictly 100% data) against these "computationally enhanced" baselines. We conducted this experiment using Gemma2-2B-IT as the full teacher and Gemma2-2B-Base as the base model, with Gemma-2B serving as the student. The results are presented below:
>
> | **Method** | Data Volum | DollyEval | SelfInst | VicunaEval | S-NI | UnNI | Avg. |
> | --- | --- | --- | --- | --- | --- | --- | --- |
> | SeqKD | 150% | 22.40 | 17.27 | 22.32 | 35.65 | 34.16 | 26.36 |
> | SelfEvol | 150% | 22.74 | 17.73 | 22.45 | 36.84 | 33.42 | 26.64 |
> | SeqKD | 100%+50% Gen | 21.80 | 17.86 | 21.91 | 35.18 | 33.08 | 25.97 |
> | SelfEvol | 100%+50% Gen | 22.34 | 17.56 | 21.73 | 36.65 | 32.29 | 26.11 |
> | Ours | 100% | **23.41** | **18.20** | **23.31** | **40.36** | **35.68** | **28.19** |
>
> As shown in the table, **DAKD consistently outperforms the best computationally-equivalent baseline**. Even when the baselines are empowered with a 50% larger data budget, they fail to match DAKD's performance. This conclusively demonstrates that the extra inference pass in DAKD is not a wasteful cost, but a highly efficient investment, which justified the computational overhead. We have included the experimental results in the Section 5 of the revised version.
>
> [1] To Repeat or Not To Repeat: Insights from Scaling LLM under Token-Crisis. NeurIPS 2023.

---

> ### Author Response · Authors · 2025-11-20
> **Response to Reviewer DXgg (3/3)**
>
> ## Q3. The paper states that " Ideally, the student would be optimized under the teacher’s true data distribution, $p^\*$... However, $p^\*$ is often inaccessible". Why is this? We have the teacher's distribution and sampling from the teacher is a standard technique.
>
> We sincerely thank the reviewer for this critical question, which allows us to clarify a core definition essential to our paper.
>
> **In short: The "distribution" the reviewer refers to and our paper's "$p^*$" are two different concepts.**
> - What the reviewer refers to (Accessible): This appears to be the teacher's "predictive distribution" $p(y|x)$ (i.e., the model's logits output).
> - What this paper refers to (Inaccessible): This is the "ideal data distribution" $p^\*(x,y)$ that was used to train the teacher model (i.e., the proprietary SFT dataset).
>
> Ideally, KD is performed on the same data distribution used to train the teacher [1,2] (e.g., DistilBERT [3] uses the original BERT corpus). However, in modern LLM scenarios (e.g., Gemma, LLaMA), the alignment data is proprietary and unreleased due to privacy and commercial restrictions [4], rendering the underlying distribution  $p^\*$
> fundamentally inaccessible. Existing work like PromptDFD [5] similarly highlights the necessity of approximating this missing target distribution.
> Our work addresses this specific challenge: we are restricted to using a generic, mismatched dataset  $\hat{p}\_{\text{data}}$, which creates a **Distributional Mismatch** relative to the ideal $p^\*$. DAKD solves this by using the Base Teacher as a probe to re-weight our accessible dataset ( $\hat{p}\_{\text{data}}$ ), thereby allowing us to better approximate the inaccessible ideal distribution ($p^\*$).
>
>
>
> We also revise Section 3 in the revised version to make the distinction between $p(y|x)$ and $p^*(x, y)$ more explicit. Furthermore, we extend the Related Work section to discuss LLMs distillation methods that address the practical unavailability of a teacher model’s original training data in the next version. We again thank the reviewer for helping us strengthen this definition.
>
> [1] Undistillable: Making a Nasty Teacher That Cannot Teach Students. ICLR 2021.
>
> [2] Small Scale Data-Free Knowledge Distillation. CVPR 2024.
>
> [3] DistilBERT, a distilled version of BERT: smaller, faster, cheaper and lighter. NeurIPS 2019.
>
> [4]Towards Zero-Shot Knowledge Distillation for Natural Language Processing. EMNLP 2021.
>
> [5] Prompting to Distill: Boosting Data-Free Knowledge Distillation via Reinforced Prompt. JCAI 2022.

---

### Official Review · Reviewer_xcEB · 2025-10-31

**Soundness:** 3
**Presentation:** 3
**Contribution:** 3
**Rating:** 6
**Confidence:** 4

**Summary:**

This paper addresses the suboptimal performance of knowledge distillation when applied to limited (and potentially suboptimal) datasets. Drawing on a similar rationale as importance sampling, the authors propose using the KL divergence between the instruction-tuned teacher model and its base version to quantify the importance of different samples/tokens/vocabulary items. Extensive empirical analysis validates the effectiveness of the proposed method.

**Strengths:**

- The proposed method appears intuitively sound and effectively addresses the key research problem of performing knowledge distillation on limited (and potentially suboptimal) datasets.

- The paper is well-structured and easily accessible to readers.

- Experimental results demonstrate superior performance compared to baseline methods, with the effectiveness of different components being systematically validated.

**Weaknesses:**

- **Potential Efficiency and Scaling Concerns**
The proposed method requires running inference with both the instruction-tuned teacher model and the base model across the entire training dataset to compute importance weights. This additional computational overhead may be significant, particularly considering that similar computational resources could be allocated to expanding the dataset size - which might naturally address the core issue of limited and suboptimal distillation data as raised in the introduction.

- **Insufficient Experimental Validation.**
The distinction between the instructed model and base model largely originates from the RLHF process designed to align the base model's outputs with human preferences (as acknowledged in the introduction). However, the experimental framework predominantly examines enhancements in reasoning capabilities, despite base models having already undergone extensive exposure to mathematical and reasoning-specific data during pretraining. This approach appears somewhat misaligned with the stated motivation. To more convincingly demonstrate the method's significance, we suggest supplementing the evaluation with experiments conducted under preference alignment settings.

- **Missing Baselines**
The study demonstrates effectiveness primarily using KL divergence. Given the rich variety of divergence measures in distillation literature (e.g., α-β-divergence [1], f-divergence [2]), it would be insightful to examine whether the observed improvements persist across different divergence formulations, thus providing a more thorough understanding of the method's robustness.

[1] ABKD: Pursuing a Proper Allocation of the Probability Mass in Knowledge Distillation via α-β-Divergence. ICML 2025

[2] f-Divergence Minimization for Sequence-Level Knowledge Distillation. ACL 2023.

**Questions:**

See Weaknesses.

---

> ### Author Response · Authors · 2025-11-20
> **Response to Reviewer xcEB (1/2)**
>
> Dear Reviewer xcEB:
>
> We appreciate your positive feedback on our work and your insightful comments.
>
> ## Q1. This additional computational overhead of the proposed model may be significant, particularly considering that similar computational resources could be allocated to expanding the dataset size.
>
> We appreciate the reviewer's insightful suggestion. We agree that we should check if the extra compute used by DAKD would be better spent on simply expanding the dataset.
>
> To test this, we ran a fair comparison:
>
> 1. **Cost Equivalence Analysis**: To determine the fair amount of additional data $\Delta N$ that equates to the computational overhead of DAKD, we derived a cost equivalence based on the parameter ratio $R = \theta_{Tea} / \theta_{Stu}$. We assume a rigorous data expansion setting that includes both Data Generation ($1 \cdot \theta_{Tea}$) and Distillation ($1 \cdot \theta_{Tea}$ for teacher logits + $3 \cdot \theta_{Stu}$ for student training), resulting in a total per-sample cost of $2\theta_{Tea} + 3\theta_{Stu}$. Equating this to DAKD's overhead ($1 \cdot \theta_{Tea}$) yields the following exchange rate formula: $\Delta N = \frac{R}{2R + 3}$.
>
> We applied this formula to the model configurations used in our paper:
>
> - Scenario A (#$\theta_{Tea}=2B$ $\to $ #$\theta_{Stu}=2B$): With $R \approx 1$, the equivalent data increase is **20%**.
> - Scenario B (#$\theta_{Tea}=9B$ $\to$ #$\theta_{Stu}=2B$): With $R \approx 4.5$, the equivalent data increase is **37.5%**.
>
> Since our work involves multiple model scales, we adopted the upper bound (37.5%) as our reference. To conduct a stress test, we rounded this up to a **50% data increase** for the baseline. This means we granted the baseline significantly more compute resources than theoretically justified, yet DAKD still outperformed it, confirming the robustness of our method.
>
> 2. **The Experiment: DAKD (100%) vs. Expanded Baseline (150%)**.
> We followed your advice and gave the baseline models **50% more data**. We implemented two data expansion strategies for the baselines (SeqKD and SelfEvol) to reach **150% data volume.**
> - 150%: Additional 50% real Data.
> - 100% + 50% Gen: Expanding data via rephrasing existing samples by teacher model.
>
> We compared DAKD (using strictly 100% data) against these "computationally enhanced" baselines. We conducted this experiment using Gemma2-2B-IT as the full teacher and Gemma2-2B-Base as the base model, with Gemma-2B serving as the student. The results are presented below:
>
> | Method | Data Volum | DollyEval | SelfInst | VicunaEval | S-NI | UnNI | Avg. |
> | --- | --- | --- | --- | --- | --- | --- | --- |
> | SeqKD | 150% | 22.40 | 17.27 | 22.32 | 35.65 | 34.16 | 26.36 |
> | SelfEvol | 150% | 22.74 | 17.73 | 22.45 | 36.84 | 33.42 | 26.64 |
> | SeqKD | 100%+50% Gen | 21.80 | 17.86 | 21.91 | 35.18 | 33.08 | 25.97 |
> | SelfEvol | 100%+50% Gen | 22.34 | 17.56 | 21.73 | 36.65 | 32.29 | 26.11 |
> | Ours | 100% | **23.41** | **18.20** | **23.31** | **40.36** | **35.68** | **28.19** |
>
> As shown in the table, **DAKD consistently outperforms the best computationally-equivalent baseline**. Even when the baselines are empowered with a 50% larger data budget, they fail to match DAKD's performance. This conclusively demonstrates that DAKD solves this by finding the right data, which turns out to be a more efficient use of compute than just adding more data. We have included the experimental results in the Section 5 of the revised version.

---

> ### Author Response · Authors · 2025-11-20
> **Response to Reviewer xcEB (2/2)**
>
> ## Q2. To more convincingly demonstrate the method's significance, we suggest supplementing the evaluation with experiments conducted under preference alignment settings.
>
> We sincerely thank the reviewer for this insightful suggestion. We fully agree that the divergence between Instruct and Base models stems from the alignment process. We respectfully submit that the enhanced reasoning performance in our original experiments also serves as valid evidence of successful alignment. Furthermore, **we conduct experiments on standard preference benchmarks** to demonstrate our method's effectiveness.
>
> 1. Recent literature establishes that the elicitation of reasoning capabilities is fundamentally a product of the post-training alignment process [1]. Consequently, reasoning benchmarks (e.g., GSM8K) are widely recognized as standard metrics for evaluating alignment effectiveness.
> Specifically, while base models acquire raw knowledge during pre-training, they lack the alignment behavior required to structure this knowledge into coherent Chain-of-Thought (CoT) sequences [2][3]. This behavioral gap represents the primary distributional discrepancy between SFT and Base models. As a result, the improved reasoning performance directly confirms that our method successfully transfers the teacher's alignment signal.
>
> 2. To further address the reviewer's suggestion and demonstrate the method's breadth, we supplemented our evaluation with AlpacaEval[4], which is the standard benchmark for preference alignment. We conducted this experiment using Gemma2-2B-IT as the full teacher and Gemma2-2B as the base model, with Gemma-2B serving as the student.
>
> | **Method** | **SFT** | **SeqKD** | **AKL** | **DAKD (Ours)** |
> | --- | --- | --- | --- | --- |
> | Win Rates | 11.64 | 11.90 | 11.27 | **14.57** |
> | Std Error | 1.15 | 1.14 | **1.12** | 1.18 |
>
> Specifically, we employed GPT-4-1106-preview as the judge to perform pairwise comparisons between each method and the teacher model to calculate the win rate. The results demonstrate that **DAKD outperforms the baseline and other comparative methods**, confirming that it effectively captures the teacher's preference alignment.
>
> We thank the reviewer for this valuable suggestion, and we have included the experimental results in the appendix of the revised version.
>
> [1] LLM post-training: A deep dive into reasoning large language models. arXiv 2025.
>
> [2] LIMA: Less Is More for Alignment. NeurIPS 2023.
>
> [3] Revisiting the Superficial Alignment Hypothesis. arXiv 2024.
>
> [4] AlpacaEval: An Automatic Evaluator of Instruction-following Models. GitHub.
>
> ## Q3. It would be insightful to examine whether the observed improvements persist across different divergence formulations (α-β-divergence, f-divergence).
>
> Thanks for your valuable suggestion. We conduct additional experiments to verify the generality of DAKD.
> DAKD functions as a data importance re-weighting framework, which is orthogonal to the specific distance metric used for loss calculation. It can be seamlessly integrated with various divergences.
>
> Following the reviewer's suggestion, we substitute the KL divergence with both the α-β-divergence and the f-Divergence. We conducted this experiment using Gemma2-2B-IT as the full teacher and Gemma2-2B-Base as the base model, with Gemma-2B serving as the student. The corresponding experimental results are presented in the table below.
>
> | **Divergence Metric** | **method** | DollyEval  | SelfInst | VicunaEval | S-NI | UnNI | Avg. |
> | --- | --- | --- | --- | --- | --- | --- | --- |
> | f-Divergence（TVD） | w/o DAKD  | 22.06 | 16.77 | 21.62 | 34.34 | 30.67 | 25.09 |
> | f-Divergence（TVD） | w/ DAKD | **24.35** | **18.98** | **22.89** | **40.35** | **32.54** | **27.82** |
> | α-β-divergence | w/o DAKD  | 22.25 | 17.55 | 22.67 | 36.01 | 30.92 | 25.88 |
> | α-β-divergence | w/ DAKD(Ours) | **23.50** | **19.37** | **23.48** | **41.94** | **33.18** | **28.30** |
>
> DAKD consistently improves performance regardless of the underlying divergence metric. This confirms that identifying and re-weighting alignment-relevant samples is a fundamental improvement that benefits various distillation objectives, verifying the method's robustness. We also include the experimental results of DAKD under these two divergence formulations in the appendix of the revised version.

---

### Official Review · Reviewer_F4Uc · 2025-10-31

**Soundness:** 2
**Presentation:** 3
**Contribution:** 2
**Rating:** 4
**Confidence:** 4

**Summary:**

Motivation of the paper: To address the problem of distribution mismatch between the high-quality knowledge in the teacher model and the knowledge learned by the student model during distillation, especially in scenarios where access to the teacher model's high-quality aligned data distribution is not available.
Method: The paper proposes a "Difference-Aware Knowledge Distillation" (DAKD) framework. Its key innovation lies in introducing a pre-trained "base version" of the teacher model as a distribution probe to measure the relevance of data to the core knowledge of the teacher.
Experiments: The paper systematically validates the effectiveness of the DAKD method in surpassing state-of-the-art approaches across various scenarios, as well as the rationality and synergy of its multi-level, dual-signal design mechanism, through comprehensive performance comparisons, data efficiency analysis, model scale evaluation, and detailed ablation studies.

**Strengths:**

1. This paper presents a clear and well-justified motivation, focusing on the important challenge of effective knowledge distillation from large to small models.
2. The paper presents a simple and intuitive method, and Section 4.1 provides some theoretical justification for the proposed approach.

**Weaknesses:**

1. The method may has certain limitations, such as the issue of cross-tokenization in real-world scenarios. and the method may be highly dependent on having access to both the base and SFT models simultaneously.
2. The method yields limited performance gains when the knowledge sources are the same.
3. The paper presents a very comprehensive quantitative analysis, It will be better that could provide a more intuitive demonstration of the method by some qualitative examples.

**Questions:**

1. If an exact base model cannot be found, or we have only to different model(Qwen/LLaMA), can the DAKD method still work? If yes, what is the performance like?
2. Regarding the hyperparameter λ that controls the "difficulty" of the learning "curriculum". A fixed λ may not be optimal. Have the authors considered annealing or adaptive strategies? For example, using a small λ in the early stages of training (resulting in a smoother weight distribution and encouraging broader exploration by the student), and gradually increasing λ as training progresses (leading to sharper weights and focusing the student on harder examples). Would this lead to more stable or better performance?

---

> ### Author Response · Authors · 2025-11-20
> **Response to Reviewer F4Uc (1/3)**
>
> Dear Reviewer F4Uc:
> Thank you for your careful reading of our paper and valuable comments.
>
> ## Q1. The method may have limitation about the issue of cross-tokenization in real-world scenarios.
>
> Overall, DAKD is a flexible and multi-level framework, and the tokenizer mismatch **only affects one of its three components (Vocabulary-level weighting)**. The other components (sequence and position-level weighting) remain valid and can provide **73%**  of the total performance gain (1.15 / 1.57). In addition, we provide **additional experiments about cross-tokenizer distillation results** (DAKD + ULD), which demonstrate that DAKD can also boost the KD results under the cross-tokenizer setting. We elaborate on these two points as follows.
>
> First, the existing ablations (Table 4 in the manuscript) quantify the contributions of the tokenizer-agnostic components. SeqKD (baseline) achieves 43.82%; using only *Sequence + Position* weighting improves accuracy to 44.97%; the full DAKD (including Vocabulary-level weighting) reaches 45.39%. Thus, full DAKD yields a +1.57% improvement (45.39 − 43.82), while the other parts alone contributes +1.15% (44.97 − 43.82). Therefore, 73% of the total gain (1.15 / 1.57) is obtained **without relying on the Vocabulary-level component**, confirming that the potential of DAKD in cross-tokenization distillation tasks.
>
> Second, we further validate DAKD **under cross-tokenizer settings** and conducted an additional experiment where the teacher and base models are Llama2-7b-chat/Llama2-7b and the student model is Gemma-2b. We use ULD [1] as the method to align the logits, which allows the DAKD to access the cross token senerio. The results provided below show that DAKD consistently enhances ULD, further supporting its generality and compatibility with existing cross-tokenizer distillation approaches. This additional experiment is also included in Section 5 in the revised version.
>
> |  | Dolly | SelfInst | Vicuna | S-NI | UnNI | Average |
> | --- | --- | --- | --- | --- | --- | --- |
> | ULD | 22.37 | 15.24 | 25.63 | 25.69 | 28.09 | 23.40 |
> | ULD+DAKD(Ours) | 22.89 | 17.82 | 26.81 | 27.10 | 31.76 | 25.28 |
>
> [1] Towards Cross-Tokenizer Distillation: the Universal Logit Distillation Loss for LLMs. TMLR 2025.
>
>
> ## Q2. The method may be highly dependent on having access to both the base and SFT models simultaneously.
>
> Our method does require both the SFT model and its corresponding base model. However, in modern open-source LLM ecosystems (e.g., Llama, Gemma, Mistral), it has become **standard practice** to release both the pretrained base model and the fine-tuned/instruct model together. This makes the base model a readily available yet often under-utilized resource that our approach naturally leverages.
>
> Furthermore, the cross-tokenizer performance demonstrated in our response to Q1 shows that the teacher and student models **do not need to belong to the same model family**, which substantially broadens the applicability of our method. Therefore, in realistic deployment scenarios, the requirement for both models does not meaningfully restrict the practical applicability of our approach.
>
> ## Q3. The method yields limited performance gains when the knowledge sources are the same.
>
> We thank the reviewer for this observation, as it highlights a key validation of our method.
>
> The reviewer correctly notes that the performance in the "same knowledge source" setting (Table 1) behaves differently than in the "different knowledge source" setting (Table 2) . This is the **expected behavior and validates our core contribution**.
> Our core contribution is that DAKD corrects distributional mismatch. The experimental setup precisely reflects this:
>
> - In the "Same Source" setting (Table 1), the mismatch is *inherently smaller* because the models share the same pre-training corpora.
> - In the "Different Source" setting (Table 2), the mismatch is *explicitly larger* because the models use different pre-training corpora .
>
> The results align with this claim that the gains are different in magnitude is not a limitation. It is a validation: it demonstrates that our method's performance correctly scales with the magnitude of the distributional mismatch it is designed to solve.
>
> Furthermore, we would like to highlight that even in this more subtle, "small-gap" setting of Table 1, our method still achieves considerable, state-of-the-art performance. DAKD (20.40) achieves a +1.39 point absolute improvement over the top-performing baseline, AKL (19.01), confirming DAKD is the most effective method in all tested scenarios.

---

> ### Author Response · Authors · 2025-11-20
> **Response to Reviewer F4Uc (2/3)**
>
> ## Q4. It will be better that could provide a more intuitive demonstration of the method by some qualitative examples.
>
> We thank the reviewer for this constructive suggestion. We agree that an intuitive demonstration helps visualize DAKD's mechanism.
>
> To demonstrate DAKD's dual-level weighting, we analyze the predictive probabilities and the resulting importance Weights on two distinct samples: a Common Knowledge sample (Capital of France) and a Multi-step Reasoning sample (Arithmetic Calculation).
>
> **Sample 1**: What is the capital of France?
>
> | Tokens | The | capital | of | France | is | Paris | Avg. |
> | --- | --- | --- | --- | --- | --- | --- | --- |
> | Base Model $p_{\texttt{base}}(y_{i}\|y_{<i})$ | 0.324 | 0.879 | 0.949 | 0.836 | 0.793 | 0.602 | **0.731** |
> | SFT Model $p_{\texttt{sft}}(y_{i}\|y_{<i})$ | 0.424 | 0.945 | 0.988 | 0.984 | 0.156 | 0.953 | **0.742** |
> | Weight | 0.81 | 0.68 | 0.60 | 1.03 | 0.02 | 2.86 | — |
>
> **Sample 2**: Calculate 8+4/2=?
>
> | Tokens | Calculate | **4** | / | 2 | = | 2 | 8 | + | 2 | = | **1** | 0 | The | Answer | is | 1 | 0 | Avg. |
> | --- | --- | --- | --- | --- | --- | --- | --- | --- | --- | --- | --- | --- | --- | --- | --- | --- | --- | --- |
> | Base Model $p_{\texttt{base}}(y_{i}\|y_{<i})$ | 0.028 | **0.278** | 0.793 | 0.992 | 0.159 | 0.871 | 0.605 | 0.941 | 0.98 | 0.578 | **0.087** | 0.996 | 0.048 | 0.017 | 0.816 | 0.957 | 0.988 | 0.596 |
> | SFT Model $p_{\texttt{sft}}(y_{i}\|y_{<i})$ | 0.34 | **0.984** | 0.992 | 1 | 0.238 | 0.879 | 1 | 0.891 | 0.992 | 0.758 | **0.965** | 1 | 0.28 | 0.025 | 0.941 | 1 | 1 | 0.781 |
> | Weight | 0.55 | **3.95** | 0.31 | 0.12 | 0.17 | 0.12 | 0.84 | 0.09 | 0.12 | 0.28 | **9.34** | 0.12 | 0.37 | 0.12 | 0.22 | 0.14 | 0.12 | — |
>
> This table provides two clear, intuitive demonstrations:
> 1. **Sequence-Level Weighting (Filtering Data):**
>
> - Sample 1 (Fact): The negligible probability gap ($\approx 0.01$) indicates the Base model already possesses this robust knowledge. Consequently, DAKD assigns a low global priority, preventing over-training on trivial facts.
> - Sample 2 (Reasoning): The significant gap signals that the Base model struggles with the logic, while the Teacher is confident. DAKD correctly identifies this as high-value alignment knowledge, triggering a **high global weight**.
>
> 2. **Position-Level:**
>
> - At token `4`(the operand), the weight jumps to **3.95** because the Base model (0.278) fails to identify the correct number to operate on, while the Teacher (0.984) is certain.
> - Most notably, at the final answer token `1` (representing "10"), the weight peaks at **9.34**. Here, the Base model (0.087) completely fails the multi-step arithmetic, while the Teacher (0.965) derives the correct result.
>
> This demonstration confirms that DAKD goes beyond simple text imitation. It automatically suppresses trivial tokens and heavily up-weights the specific reasoning steps (e.g., operands and results) where the teacher's alignment expertise provides the most critical correction to the base model.

---

> ### Author Response · Authors · 2025-11-20
> **Response to Reviewer F4Uc (3/3)**
>
> ## Q5. If an exact base model cannot be found, or we have only to different model(Qwen/LLaMA), can the DAKD method still work? If yes, what is the performance like?
>
> Thanks for your thoughtful question. While open-source models are always released in pairs, we agree that cross-architecture flexibility is worth considering. We perform a stress test in which the “probe” (base) model’s architecture is different from that of the SFT teacher model. Specifically, we conduct an experiment using Gemma-2-9B-IT as the teacher and Llama-2-7B as the mismatched “probe”(base), and keep the student as Gemma-2B. For the token and logits mismatch introduced by different tokenizer between teacher and base models, we utilize ULD [1] to align both token and logits. We also introduce a Sequence+Position level weights with Dynamic Time Warping (DTW) [2] to align the token. The experimental results are shown in the table below.
>
> |  | Teacher Model | Base Model | Alignment | Dolly | SelfInst | Vicuna | S-NI | UnNI | Average |
> | --- | --- | --- | --- | --- | --- | --- | --- | --- | --- |
> | SeqKD | Gemma-2-9B-IT | NA | NA | 22.4 | 17.27 | **22.32** | 35.67 | 34.16 | 26.36 |
> | DAKD | Gemma-2-9B-IT | Llama-2-7B | ULD [1] | 22.51 | 17.08 | 21.2 | 34.65 | 33.47 | 25.78 |
> | DAKD | Gemma-2-9B-IT | Llama-2-7B | DTW [2] | **23.01** | **18.28** | 21.37 | **36.62** | **34.31** | **26.72** |
>
> The results show that the ULD version did not achieve the baseline level. This may be attributed to the mismatch caused by heterogeneous tokenizers, which introduces excessive noise into the weights. To minimize this noise, we discarded the logits-level weighting and relied solely on Sequence- and Position-level weights, employing DTW for token-level alignment. Consequently, this adapted approach outperforms the baseline, but underperforms our methods with matched base model. The performance gap between the mismatched and matched scenarios highlights that a homologous base model acts as a much more precise probe. We recommend using the homologous base model whenever possible.
>
> [1] Towards cross-tokenizer distillation: the universal logit distillation loss for LLMs. TMLR 2025.
>
> [2] Specializing Smaller Language Models towards Multi-Step Reasoning. PMLR 2023.
>
> ## Q6. Consider annealing strategy for hyperparameter $\lambda$.
>
> Thank you for this practical and insightful suggestion. Following your recommendation, we conducte a new experiment to evaluate it. We implement a linear warm-up schedule for $\lambda$, increasing from 0 to 5 over the first 50% of training steps, and remaining fixed at 5 thereafter. We conducted this experiment using Gemma2-2B-IT as the full teacher and Gemma2-2B as the base model, with Gemma-2B serving as the student.
>
> | **Method** | **GSM8K** | **DROP** | **Hellaswag** | **Average** |
> | --- | --- | --- | --- | --- |
> | Fixed $\lambda=5$ | **45.39** | **40.83** | **55.95** | **47.39** |
> | Annealing  $\lambda$  | 45.05 | 40.52 | 55.58 | 47.05 |
>
> We observed that the annealed $\lambda$ yielded performance highly comparable to, though not significantly exceeding, our optimal fixed $\lambda$. It indicates that while adaptive strategies are a valid design direction, DAKD is robust to the specific scheduling of $\lambda$ and performs effectively with a constant value.
>
> We thank the reviewer for this constructive suggestion, which has helped verify the method's stability, and we also include this discussion in the appendix of the revised version.

---

### Author Response · Authors · 2025-11-20
**General Response to All Reviewers**

We sincerely appreciate the reviewers' time and the valuable feedback they have provided for our paper. These constructive comments have been instrumental in enhancing the quality of our work. We are encouraged that reviewers recognized our method as “intuitively sound” (Reviewers xcEB and F4Uc), having “clear and well-justified motivation” (Reviewer F4Uc), and showing “superior experimental performance” (Reviewers xcEB and DXgg).

Below, we provide point-by-point responses to your comments and present the revisions made to the manuscript based on your suggestions. Notably, a majority of the comments recommend adding further experiments. In response, we have conducted a comprehensive set of new experiments, which we summarize below:

- New LLM Base: Heterogeneous models, and cross-tokenizer alignment.
- Qualitative Analysis: Visualization results.
- Hyperparameter Strategy: Annealing schedule for the parameter $\lambda$.
- Computational Cost Analysis: Comparison against baselines with 50% expanded data volume.
- New Benchmark: AlpacaEval for preference alignment settings.
- New Objectives: $\alpha$-$\beta$-divergence and $f$-divergence.

All revisions are highlighted in blue. We hope that our responses and additional experiments could address your concerns.

---

### Author Response · Authors · 2025-11-30
**Summary of the Rebuttal for the Area Chair**

Dear Area Chair,

We sincerely appreciate your time and attention and thank the reviewers for their thoughtful comments.

In the first week of rebuttal, we had already submitted both the full rebuttal and the revised version. Unfortunately, we have not received any further responses from the reviewers, which limits the possibility of additional discussion or clarification. **We kindly ask you to fully consider our rebuttal and the additional experimental results when assessing whether the reviewers’ concerns have been adequately resolved.**

In light of this, we briefly summarize the reviewers’ assessments and how we addressed the reviewers’ concerns.

We are encouraged that the reviewers generally view the paper positively in terms of motivation, theory, intuitiveness, and experimental performance:

- **Clear motivation.** Reviewer F4Uc describes the work as having “a clear and well-justified motivation, focusing on the important challenge” while reviewer xcEB notes that the paper “effectively addresses the key research problem.”
- **Theoretical analysis.** Reviewer F4Uc acknowledges that the paper “provides some theoretical justification” for the proposed method.
- **Intuitive appeal.** Reviewer F4Uc highlights that our approach is “simple and intuitive,” and reviewer xcEB considers it “intuitively sound.”
- **Experimental performance.** Reviewer xcEB recognizes the “superior performance compared to baseline methods,” and reviewer DXgg notes that the method “shows improvements over existing distillation methods.”

We have tried our best to address all questions and concerns raised by the reviewers. We have conducted additional experiments, highlighted key experimental results in the paper, and provided clarifications to resolve any potential misunderstandings.

- A shared concern raised by Reviewers xcEB and DXgg regards the justification for the **additional computational overhead** introduced by the base model. By comparing our approach with data augmentation method (**Q1 for Reviewer xcEB** & **Q2 for Reviewer DXgg**), we demonstrate that **our method represents a more efficient use of compute**.
- To address **practicality concerns** (Reviewer F4Uc), we demonstrated the effectiveness of our method in **cross-vocabulary scenarios** (**Q1 for Reviewer F4Uc**) and provided a robust solution for **teacher-base mismatch** (**Q5 for Reviewer F4Uc**) even outside the primary experimental setting. We also clarified the **high prevalence of open-source base models** in white-box distillation settings (**Q2 for Reviewer F4Uc**). These results and clarifications further demonstrate the **broad applicability** of the proposed method **in real-world scenarios**.
- Following the suggestions from Reviewer xcEB, we have extended our validation to include **additional baselines and new tasks** (**Q3 & Q2 for Reviewer xcEB**). Specifically, we incorporated experiments using two new distillation loss functions ($\alpha$-$\beta$ divergence and $f$-divergence) and evaluated performance on the preference alignment benchmark, AlpacaEval.
- Additionally, we have provided:
    - **Visualization results** to intuitively demonstrate the mechanism of the proposed method (**Q4 for Reviewer F4Uc**).
    - Extended experiments and analysis on **hyperparameters** (**Q6 for Reviewer F4Uc** & **Q1 for Reviewer DXgg**), which further confirm the robustness of our approach.
    - Clarifications on specific technical details to resolve misunderstandings (**Q3 for Reviewer F4Uc** & **Q3 for DXgg**).

We have provided a detailed, **point-by-point response** to each of the reviewers' concerns. We are sincerely grateful for the constructive suggestions that guided our improvements. **We have updated the manuscript** with the new experiments, emphasized findings, and clarifications. We hope this work will contribute to the AIGC community, not only by presenting an effective method but also by demonstrating the potential of Knowledge Distillation and **promoting the efficient utilization of open-source resources**.

Last, we provide the summerization of our manuscript. We identify a critical distribution mismatch issue that arises in the distillation of LLMs. By leveraging the often-overlooked **base model**, we propose a three-level re-weighting mechanism guide the student model’s attention, thereby improving its learning effectiveness. We further show clear improvements on instruction following, reasoning, and preference alignment (added during rebuttal) tasks.

Once again, we sincerely appreciate your time and effort. **If you have any further concerns regarding our work or our rebuttal, we would be more than happy to provide additional responses and continue the discussion.**

Sincerely,

Authors of Paper 11759

---

### Meta-Review · Area_Chair_d9eq · 2026-01-07

**Summary:**

This paper proposes **DAKD (Discrepancy-Aware Knowledge Distillation)**, which uses discrepancy signals (teacher–student and teacher–base) to **reweight distillation** at the sequence, position, and vocabulary levels, aiming to mitigate distribution mismatch in LLM distillation. The paper is clearly written, and the rebuttal added helpful ablations and qualitative examples.

However, several core issues remain insufficiently resolved:

1. **Limited conceptual novelty**:

The method is still perceived as a combination of established distillation ideas (importance reweighting / curriculum-style weighting), with discrepancy used as an additional signal. The rebuttal improves completeness but does not change the overall impression that the conceptual step is incremental.

2. **Weak theory–algorithm connection**:

The theory does not clearly compel the specific design choices (two discrepancy sources, the exact weighting form, and their combination). The added sensitivity analyses help, but the argument remains more motivational than determinative.

3. **Practical overhead and assumptions**:

DAKD introduces non-trivial overhead (e.g., extra model passes and reliance on base-teacher availability), and performance may depend on compatibility assumptions (e.g., tokenizer/model pairing). The compute-equivalent discussion is useful, but the cost–benefit tradeoff remains uncertain in realistic settings.

Given these concerns, the current evidence is not sufficient for acceptance. I encourage the authors to clarify what is fundamentally new, strengthen the theory-to-method linkage, and broaden evaluation under realistic constraints.

**Reviewer Concerns:**

**Addressed by the rebuttal**:

- Added ablations (e.g., α sensitivity, annealing) and clearer qualitative explanations.
- Added comparisons intended to control for extra compute.

**Unsolved Issues**:

- Novelty remains perceived as incremental.
- The theory does not clearly determine the algorithmic form.
- Overhead and dependency assumptions remain concerns for practical use.

**Reviewer Scores:**

- **Reviewer F4Uc**: Core concerns likely remain (generality under cross-tokenizer/non-matched settings; incremental framing). The score should be kept unchanged.
- **Reviewer xcEB**: Key concerns likely remain (overhead vs benefit; theory not compelling the specific weighting design).  The score should be kept unchanged.
- **Reviewer DXgg**: Main concerns likely remain (incremental novelty; weak theory-to-algorithm link).  The score should be kept unchanged.

---

### Decision · Program_Chairs · 2026-01-26

Reject